# Understanding Image and Text Simultaneously: a Dual Vision-Language Machine Comprehension Task

## Abstract

We introduce a new multi-modal task for computer systems, posed as a combined vision-language comprehension challenge: identifying the most suitable *text* describing a scene, given several similar options. Accomplishing the task entails demonstrating comprehension beyond just recognizing "keywords" (or key-phrases) and their corresponding visual concepts. Instead, it requires an alignment between the representations of the two modalities that achieves a visually-grounded "understanding" of various linguistic elements and their dependencies. This new task also admits an easy-to-compute and well-studied metric: the accuracy in detecting the true target among the decoys.

The paper makes several contributions: an effective and extensible mechanism for generating decoys from (human-created) image captions; an instance of applying this mechanism, yielding a large-scale machine comprehension dataset (based on the COCO images and captions) that we make publicly available; human evaluation results on this dataset, informing a performance upper-bound; and several baseline and competitive learning approaches that illustrate the utility of the proposed task and dataset in advancing both image and language comprehension. We also show that, in a multi-task learning setting, the performance on the proposed task is positively correlated with the end-to-end task of image captioning.

## 1 Introduction

There has been a great deal of interest in multi-modal artificial intelligence research recently, bringing together the fields of Computer Vision and Natural Language Processing. This interest has been fueled in part by the availability of many large-scale image datasets with textual annotations. Several vision+language tasks have been proposed around these datasets (Hodosh et al., 2013; Karpathy and Fei-Fei, 2015; Lin et al., 2014; Antol et al., 2015). Image Captioning (Hodosh et al., 2013; Donahue et al., 2014; Karpathy and Fei-Fei, 2015; Fang et al., 2015; Kiros et al., 2015; Vinyals et al., 2015; Mao et al., 2015; Xu et al., 2015b) and Visual Question Answering (Malinowski and Fritz, 2014; Malinowski et al., 2015; Tu et al., 2014; Antol et al., 2015; Yu et al., 2015; Wu et al., 2016b; Ren et al., 2015; Gao et al., 2015; Yang et al., 2016; Zhu et al., 2016; Lin and Parikh, 2016) have in particular attracted a lot of attention. The performances on these tasks have been steadily improving, owing much to the wide use of deep learning architectures (Bengio, 2009).

A central theme underlying these efforts is the use of natural language to identify how much visual information is perceived and understood by a computer system. Presumably, a system that understands a visual scene well enough ought to be able to describe what the scene is about (thus "captioning") or provide correct and visually-grounded answers when queried (thus "question-answering").

In this paper, we argue for directly measuring how well the semantic representations of the visual and linguistic modalities align (in some abstract semantic space). For instance, given an image and two captions – a correct one and an incorrect yet-cunningly-similar one – can we both qualitatively and quantitatively measure the ex-

tent to which humans can dismiss the incorrect one but computer systems blunder? Arguably, the degree of the modal alignment is a strong indicator of task-specific performance on any vision+language task. Consequentially, computer systems that can learn to maximize and exploit such alignment should outperform those that do not.

We take a two-pronged approach for addressing this issue. First, we introduce a new and challenging Dual Machine Comprehension (DMC) task, in which a computer system must identify the most suitable textual description from several options: one being the target and the others being "adversarialy"-chosen decoys. All options are free-form, coherent, and fluent sentences with *high degrees of semantic similarity* (hence, they are "cunningly similar"). A successful computer system has to demonstrate comprehension beyond just recognizing "keywords" (or key phrases) and their corresponding visual concepts; they must arrive at a coinciding and visually-grounded understanding of various linguistic elements and their dependencies. What makes the DMC task even more appealing is that it admits an easy-to-compute and well-studied performance metric: the accuracy in detecting the true target among the decoys. Second, we illustrate how solving the DMC task benefits related vision+language tasks. To this end, we render the DMC task as a classification problem, and incorporate it in a multi-task learning framework for end-to-end training of joint objectives.

Our work makes the following contributions: (1) an effective and extensible algorithm for generating decoys from human-created image captions (Section 3.2); (2) an instantiation of applying this algorithm to the COCO dataset (Lin et al., 2014), resulting in a large-scale dual machine-comprehension dataset that we make publicly available (Section 3.3); (3) a human evaluation on this dataset, which provides an upper-bound on performance (Section 3.4); (4) a benchmark study of baseline and competitive learning approaches (Section 5), which underperform humans by a substantial gap (about 20%); and (5) a multi-task learning model that simultaneously learns to solve the DMC task and the Image Captioning task (Section 4).

Our empirical study shows that performance on the DMC task positively correlates with performance on the Image Captioning task. Therefore, besides acting as a standalone benchmark, the new DMC task can be useful in improving other complex vision+language tasks. Both suggest the DMC task as a fruitful direction for future research.

## 2 Related work

Image understanding is a long-standing challenge in computer vision. There has recently been a great deal of interest in bringing together vision and language understanding. Particularly relevant to our work are image captioning (IC) and visual question-answering (VQA). Both have instigated a large body of publications, a detailed exposition of which is beyond the scope of this paper. Interested readers should refer to two recent surveys (Bernardi et al., 2016; Wu et al., 2016a).

In IC tasks, systems attempt to generate a fluent and correct sentence describing an input image. IC systems are usually evaluated on how well the generated descriptions align with human-created captions (ground-truth). The language generation model of an IC system plays a crucial role; it is often trained such that the probabilities of the ground-truth captions are maximized (MLE training), though more advanced methods based on techniques borrowed from Reinforcement Learning have been proposed (Ranzato et al., 2015). To provide visual grounding, image features are extracted and injected into the language model. Note that language generation models need to both decipher the information encoded in the visual features, and model natural language generation.

In VQA tasks, the aim is to answer an input question correctly with respect to a given input image. In many variations of this task, answers are limited to single words or a binary response ("yes" or "no") (Antol et al., 2015). The Visual7W dataset (Zhu et al., 2016) contains anaswers in a richer format such as phrases, but limits questions to "wh-"style (what, where, who, etc). The Visual Genome dataset (Krishna et al., 2016), on the other hand, can potentially define more complex questions and answers due to its extensive textual annotations.

Our DMC task is related but significantly different. In our task, systems attempt to discriminate the best caption for an input image from a set of captions — all but one are decoys. Arguably, it is a form of VQA task, where the same default

(thus uninformative) question is asked: *Which of the following sentences best describes this image?* However, unlike current VQA tasks, choosing the correct answer in our task entails a deeper "understanding" of the available answers. Thus, to perform well, a computer system needs to understand both complex scenes (visual understanding) and complex sentences (language understanding), *and* be able to reconcile them.

The DMC task admits a simple classification-based evaluation metric: the accuracy of selecting the true target. This is a clear advantage over the IC tasks, which often rely on imperfect metrics such as BLEU (Papineni et al., 2002), ROUGE (Lin and Och, 2004), METEOR (Banerjee and Lavie, 2005), CIDEr (Vedantam et al., 2015), or SPICE (Anderson et al., 2016).

Related to our proposal is the work in (Hodosh et al., 2013), which frames image captioning as a ranking problem. While both share the idea of selecting captions from a large set, our framework has some important and distinctive components. First, we devise an algorithm for smart selection of candidate decoys, with the goal of selecting those that are sufficiently similar to the true targets to be challenging, and yet still be reliably identifiable by human raters. Second, we have conducted a thorough human evaluation in order to establish a performance ceiling, while also quantifying the level to which current learning systems underperform. Lastly, we show that there exists a positive correlation between the performance on the DMC task and the performance on related vision+language tasks by proposing and experimenting with a multi-task learning model. Our work is also substantially different from their more recent work (Hodosh and Hockenmaier, 2016), where only one decoy is considered and its generation is either random, or focusing on visual concept similarity ("switching people or scenes") instead of our focus on both linguistic surface and paragraph vector embedding similarity.

## 3 The Dual Machine Comprehension Task

### 3.1 Design overview

We propose a new multi-modal machine comprehension task to examine how well visual and textual semantic understanding are aligned. Given an image, human evaluators or machines must accurately identify the best sentence describing the

scene from several decoy sentences. Accuracy on this task is defined as the percentage that the true targets are identified.

It seems straightforward to construct a dataset for this task, as there are several existing datasets which are composed of images and their (multiple) ground-truth captions, including the popular COCO dataset (Lin et al., 2014). Thus, for any given image, it appears that one just needs to use the captions corresponding to other images as decoys. However, this naïve approach could be overly simplistic as it is provides no control over the properties of the decoys.

Specifically, our desideratum is to recruit *challenging* decoys that are sufficiently similar to the targets. However, for a small number of decoys, e.g. 4-5, randomly selected captions could be significantly different from the target. The resulting dataset would be too "easy" to shed any insight on the task. Since we are also interested in human performance on this task, it is thus impractical to increase the number of decoys to raise the difficulty level of the task at the expense of demanding humans to examine tediously and unreliably a large number of decoys. In short, we need an *automatic procedure to reliably create difficult sets of decoy captions* that are sufficiently similar to the targets.

We describe such a procedure in the following. While it focuses on identifying decoy captions, the main idea is potentially adaptable to other settings. The algorithm is flexible in that the "difficulty" of the dataset can be controlled to some extent through the algorithm's parameters.

### 3.2 Algorithm to create an MC-IC dataset

The main idea behind our algorithm is to carefully define a "good decoy". The algorithm exploits recent advances in paragraph vector (PV) models (Le and Mikolov, 2014), while also using linguistic surface analysis to define similarity between two sentences. Due to space limits, we omit a detailed introduction of the PV model. It suffices to note that the model outputs a continuously-valued embedding for a sentence, a paragraph, or even a document.

The pseudo-code for the algorithm is in the Listing 1 (the name MC-IC stands for "Machine-Comprehension for Image-Captions"). As input, the algorithm takes a large set $C$ of

**Algorithm 1:** MC-IC($C$, $N$, $Score$)

**Result**: Dataset $MC_{IC}$
$PV \leftarrow$ OPTIMIZE-PV($C$)
$\lambda \leftarrow$ OPTIMIZE-SCORE($PV$, $C$, $Score$)
$MC_{IC} \leftarrow \emptyset$
$nr\_decoys = 4$
**for** $\langle \mathbf{i}_i, \mathbf{c}_i \rangle \in C$ **do**
 $A \leftarrow []$
 $T_{\mathbf{c}_i} \leftarrow PV(\mathbf{c}_i)[1..N]$
 **for** $\mathbf{c}_d \in T_{\mathbf{c}_i}$ **do**
 $score \leftarrow Score(PV, \lambda, \mathbf{c}_d, \mathbf{c}_i)$
 **if** $score > 0$ **then**
 $A$.**append**($\langle score, \mathbf{c}_d \rangle$)
 **end**
 **end**
 **if** $|A| \geq nr\_decoys$ **then**
 $R \leftarrow$ **descending-sort**($A$)
 **for** $l \in [1..nr\_decoys]$ **do**
 $\langle score, \mathbf{c}_d \rangle \leftarrow R[l]$
 $MC_{IC} \leftarrow MC_{IC} \cup \{(\langle \mathbf{i}_i, \mathbf{c}_d \rangle, \textbf{false})\}$
 **end**
 $MC_{IC} \leftarrow MC_{IC} \cup \{(\langle \mathbf{i}_i, \mathbf{c}_i \rangle, \textbf{true})\}$
 **end**
**end**

$\langle image, \{caption(s)\} \rangle$ pairs[*], as those extracted from a variety of publicly-available corpora, including the COCO dataset (Lin et al., 2014). The output of the algorithm is the dataset $MC_{IC}$ which is used for the DMC task.

Concretely, the MC-IC Algorithm has three main arguments: a dataset $C = \{\langle \mathbf{i}_i, \mathbf{c}_i \rangle | 1 \leq i \leq m\}$ where $\mathbf{i}_i$ is an image and $\mathbf{c}_i$ is its ground-truth caption. For an image with multiple groundtruth captions, we split it to multiple instances with the same image and one unique groundtruth caption per instance. An integer $N$ which controls the size of $\mathbf{c}_i$'s neighborhood in the embedding space defined by the paragraph vector model $PV$; and a function $Score$ which is used to score the $N$ items in each such neighborhood.

The first two steps of the algorithm tune several hyperparameters. The first step finds optimal settings for the $PV$ model given the dataset $C$. The second finds a weight parameter $\lambda$ given $PV$, dataset $C$, and the $Score$ function. These hyperparameters are dataset-specific. Details are discussed in the next section.

The main body of the algorithm, the outer **for** loop, generates a set of $nr\_decoys$ (4 here) decoys for each ground-truth caption. It accomplishes this by first extracting $N$ candidates from the $PV$ neighborhood of the ground-truth caption, excluding those that belong to the same image.

---

In the inner **for** loop, it computes the similarity of each candidate to the ground-truth and stores them in a list $A$. If enough candidates are generated, the list is sorted in descending order of score. The top $nr\_decoys$ captions are marked as "decoys" (i.e. **false**), while the ground-truth caption is marked as "target" (i.e. **true**).

The score function $Score(PV, \lambda, \mathbf{c}', \mathbf{c})$ is a crucial component of the decoy selection mechanism. Its definition leverages our linguistic intuition by combining linguistic surface similarity, $\mathsf{sim}_{\mathrm{SURF}}(\mathbf{c}', \mathbf{c})$, with the similarity suggested by the embedding model, $\mathsf{sim}_{\mathrm{PV}}(\mathbf{c}', \mathbf{c})$:

$$\mathsf{Score} = \begin{cases} 0 & \text{if } \mathsf{sim}_{\mathrm{SURF}} \geq L \\ \lambda\,\mathsf{sim}_{\mathrm{PV}} + (1-\lambda)\,\mathsf{sim}_{\mathrm{SURF}} & \text{otherwise} \end{cases} \quad (1)$$

where the common argument $(\mathbf{c}', \mathbf{c})$ is omitted. The higher the similarity score, the more likely that $\mathbf{c}'$ is a good decoy for $\mathbf{c}$. Note that if the surface similarity is above the threshold $L$, the function returns 0, flagging that the two captions are too similar to be used as a pair of target and decoy.

In this work, $\mathsf{sim}_{\mathrm{SURF}}$ is computed as the BLEU score between the inputs (Papineni et al., 2002) (with the brevity penalty set to 1). The embedding similarity, $\mathsf{sim}_{\mathrm{PV}}$, is computed as the cosine similarity between the two in the PV embedding space.

### 3.3 The $MC_{IC}$ dataset

We applied the MC-IC Algorithm to the COCO dataset (Lin et al., 2014) to generate a dataset for the visual-language dual machine comprehension task. The dataset, called $MC_{IC}$ , is made publicly available (address anonymized). We describe the details of this dataset below.

We set the neighborhood size at $N = 500$, and the threshold at $L = 0.5$ (see Eq. 1). As the COCO dataset has a large body of images (thus captions) focusing on a few categories (such as sports activities), this threshold is important in discarding significantly similar captions to be decoys – otherwise, even human annotators will experience difficulty in selecting the ground-truth captions.

The hyperparameters of the $PV$ model, dim (embedding dimension) and epochs (number of training epochs), are optimized in the OPTIMIZE-PV step of the MC-IC Algorithm. The main idea is to learn embeddings such that ground-truth captions from the same image have similar embeddings. Details are in the Suppl. Ma-

| Split | dev | test | train | total |
|---|---|---|---|---|
| #unique˙images | 2,000 | 2,000 | 110,800 | 114,800 |
| # instances | 9,999 | 10,253 | 554,063 | 574,315 |

Table 1: MC$_{\text{IC}}$ dataset descriptive statistics

terial. The optimal settings are `dim`=1024 and `epochs`=5.

Likewise, the $\lambda$ in the Score function is optimized by the OPTIMIZE-SCORE step, such that ground-truth captions associated with the same image have similar scores. Details are in the Suppl. Material. The optimal $\lambda$ is 0.3.

The resulting MC$_{\text{IC}}$ dataset has 574,315 instances that are in the format of $\{i : (\langle \mathbf{i}_i, \mathbf{c}_i^j \rangle, \text{label}_i^j), j = 1 \dots 5\}$ where $\text{label}_i^j \in \{\textbf{true}, \textbf{false}\}$. For each such instance, there is one and only one $j$ such that the label is **true**. We have created a train/dev/test split such that all of the instances for the same image occur in the same split. Table 1 reports the basic statistics for the dataset. Figure 1 shows several instances from the MC$_{\text{IC}}$ dataset.

### 3.4 Human performance on MC$_{\text{IC}}$

To measure how well humans can perform on the DMC task, we randomly drew 1,000 instances from the MC$_{\text{IC}}$ dev set and submitted those instances to human "raters"[†] via a crowdsourcing platform.

Three independent responses from 3 different raters were gathered for each instance, for a total of 3,000 responses. To ensure diversity, raters were prohibited from evaluating more than six instances or from responding to the same task instance twice. In total, 807 distinct raters were employed.

Raters were shown one instance at a time. They were shown the image and the five caption choices (ground-truth and four decoys) and were instructed to choose the best caption for the image. To supplement the instructions, raters were initially shown a few examples from the training set with the ground-truth caption highlighted, to illustrate how to discern the most appropriate caption for the image (see the Suppl. Material for details).

We assessed human performance using two metrics: (1) Percentage of correct rater responses (1-human system): **81.1%** (2432 out of 3000); (2)

---

[†]Raters are vetted, screened and tested before working on any tasks; requirements include native-language proficiency level.

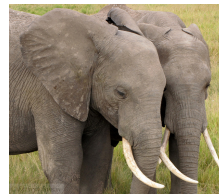

1. a herd of giraffe standing next to each other in a dirt field
2. a pack of elephants standing next to each other
3. animals are gathering next to each other in a dirt field
4. three giraffe standing next to each other on a grass field
5. **two elephants standing next to each other in a grass field**

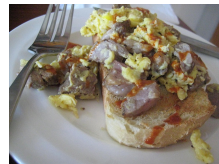

1. a meal covered with a lot of broccoli and tomatoes
2. a pan filled with a mixture of vegetables and meat
3. **a piece of bread covered in a meat and sauces**
4. a pizza smothered in cheese and meat with french fries
5. a plate of fries and a sandwich cut in half

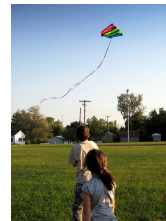

1. a family is playing the wii in a house
2. a man playing with a frisbee in a park
3. a small boy playing with kites in a field
4. **boy flies a kite with family in the park**
5. three women play with frisbees in a shady park

Figure 1: Examples of instances from the MC$_{\text{IC}}$ dataset (the ground-truth is in **bold** face). The third example illustrates one of the difficult cases in which the humans annotators did not agree (option 3. was also chosen).

Percentage of instances with at least $50\%$ (i.e. 2) correct responses (3-human system): **82.8%** (828 out of 1000). Due to space limitation, more discussions about the human raters disagreement is available in the Suppl. Material.

## 4 Learning Methods

We describe several learning methods for the dual machine comprehension (DMC) task with the MC$_{\text{IC}}$ dataset.

**Regression.** To examine how well the two embeddings are aligned in "semantic understanding space", a simple approach is to assume that the learners do not have access to the decoys. Instead, by accessing the ground-truth captions only,

the models learn a linear regressor from the image embeddings to the target captions' embeddings ("forward regression"), or from the captions to the images ("backward regression"). With the former approach, referred as Baseline-I2C, we check whether the predicted caption for any given image is closest to its true caption. With the latter, referred as Baseline-C2I, we check whether the predicted image embedding by the ground-truth caption is the closest among predicted ones by decoy captions to the real image embeddings.

**Linear classifier.** Our next approach Baseline-LinM is a linear classifier learned to discriminate true targets from the decoys. Specifically, we learn a linear discriminant function $f(\mathbf{i}, \mathbf{c}; \boldsymbol{\theta}) = \mathbf{i}^\top \boldsymbol{\theta} \, \mathbf{c}$ where $\boldsymbol{\theta}$ is a matrix measuring the compatibility between two types of embeddings, cf. (Frome et al., 2013). The loss function is then given by

$$L(\boldsymbol{\theta}) = \sum_i [\max_{j \neq j^*} f(\mathbf{i}_i, \mathbf{c}_i^j; \boldsymbol{\theta}) - f(\mathbf{i}_i, \mathbf{c}_i^{j^*}; \boldsymbol{\theta})]_+ \quad (2)$$

where $[\,]_+$ is the hinge function and $j$ indexes over all the available decoys and $i$ indexes over all training instances. The optimization tries to increase the gap between the target $\mathbf{c}_i^{j^*}$ and the worst "offending" decoy. We use stochastic (sub)gradient methods to optimize $\boldsymbol{\theta}$, and select the best model in terms of accuracy on the $\mathrm{MC_{IC}}$ dev set.

**FFNN Model.** To present our neural-network–based models, we use the following notations. Each training instance pair is a tuple $\langle \mathbf{i}_i, \mathbf{c}_i^j \rangle$, where $\mathbf{i}$ denotes the image, and $\mathbf{c}_i^j$ denotes the caption options, which can either be the target or the decoys. We use a binary variable $y_{ijk} \in \{0, 1\}$ to denote whether $j$-th caption of the instance $i$ is labeled as $k$, and $\sum_k y_{ijk} = 1$.

We first employ the standard feedforward neural-network models to solve the $\mathrm{MC_{IC}}$ task. For each instance pair $\langle \mathbf{i}_i, \mathbf{c}_i^j \rangle$, the input to the neural network is an embedding tuple $\langle \mathrm{DNN}(\mathbf{i}_i; \Gamma), \mathrm{Emb}(\mathbf{c}_i^j; \Omega) \rangle$, where $\Gamma$ denotes the parameters of a deep convolutional neural network DNN. DNN takes an image and outputs an image embedding vector. $\Omega$ is the embedding matrix, and $\mathrm{Emb}(.)$ denotes the mapping from a list of word IDs to a list of embedding vectors using $\Omega$. The loss function for our FFNN is given by:

$$L(\Gamma, \Omega, \mathbf{u}) = \sum_{i,j,k} y_{ijk} \log \, \mathrm{FN}_k(\mathrm{DNN}(\mathbf{i}_i; \Gamma), \mathrm{Emb}(\mathbf{c}_i^j; \Omega); \mathbf{u})$$

$$(3)$$

where $\mathrm{FN}_k$ denotes the $k$-th output of a feedforward neural network, and $\sum_k \mathrm{FN}_k(.) = 1$. Our architecture uses a two hidden-layer fully connected network with Rectified Linear hidden units, and a softmax layer on top.

The formula in Eq. 3 is generic with respect to the number of classes. In particular, we consider a 2-class–classifier ($k \in \{0, 1\}$, 1 for 'yes', this is a correct answer; 0 for 'no', this is an incorrect answer), applied independently on all the $\langle \mathbf{i}_i, \mathbf{c}_i^j \rangle$ pairs and apply one FFNN-based binary classifier for each; the final prediction is the caption with the highest 'yes' probability among all instance pairs belonging to instance $i$.

**Vec2seq + FFNN Model.** We describe here a hybrid neural-network model that combines a recurrent neural-network with a feedforward one. We encode the image into a single-cell RNN encoder, and the caption into an RNN decoder. Because the first sequence only contains one cell, we call this model a vector-to-sequence (Vec2seq) model as a special case of Seq2seq model as in (Sutskever et al., 2014; Bahdanau et al., 2015). The output of each unit cell of a Vec2seq model (both on the encoding side and the decoding side) can be fed into an FFNN architecture for binary classification (see the Suppl. Material for an architecture illustration).

In addition to the classification loss (Eq. 3), we also include a loss for generating an output sequence $\mathbf{c}_i^j$ based on an input $\mathbf{i}_i$ image. We define a binary variable $z_{ijlv} \in \{0, 1\}$ to indicate whether the $l$th word of $\mathbf{c}_i^j$ is equal to word $v$. $\mathbf{O}_{ijl}^d$ denotes the $l$-th output of the decoder of instance pair $\langle \mathbf{i}_i, \mathbf{c}_i^j \rangle$, $\mathbf{O}_{ij}^e$ denotes the output of the encoder, and $\mathbf{O}_{ij:}^d$ denotes the concatenation of decoder outputs.

With these definitions, the loss function for the Vec2seq + FFNN model is:

$$L(\boldsymbol{\theta}, \mathbf{w}, \mathbf{u})$$
$$= \sum_{i,j,k} y_{ijk} \log \, \mathrm{FN}_k(\mathbf{O}_{ij}^e(\mathbf{i}_i, \mathbf{c}_i^j; \boldsymbol{\theta}), \mathbf{O}_{ij:}^d(\mathbf{i}_i, \mathbf{c}_i^j; \boldsymbol{\theta}); \mathbf{u})$$
$$+ \lambda_{gen} \sum_{i,j,l,v} y_{ij1} z_{ijlv} \log \, \mathrm{softmax}_v(\mathbf{O}_{ijl}^d(\mathbf{i}_i, \mathbf{c}_i^j; \boldsymbol{\theta}); \mathbf{w})$$
$$(4)$$

where $\sum_v \mathrm{softmax}_v(.) = 1$; $\boldsymbol{\theta}$ are the parameters of the Vec2seq model, which include the parameters within each unit cell, as well as the elements in the embedding matrices for images and target sequences; $\mathbf{w}$ are the output projection parameters that transform the output space of the decoder to the vocabulary space. $\mathbf{u}$ are the parameters of the

FFNN model (Eq. 3); $\lambda_{gen}$ is the weight assigned to the sequence-to-sequence generation loss. Only the true target candidates (the ones with $y_{ij1} = 1$) are included in this loss, as we do not want the decoy target options to affect this computation.

The Vec2seq model we use here is an instantiation of the attention-enhanced models proposed in (Bahdanau et al., 2015; Chen et al., 2016). However, our current model does not support location-wise attention, as in the Show-Attend-and-Tell (Xu et al., 2015a) model. In this sense, our model is an extension of the Show-and-Tell model with a single attention state representing the entire image, used as image memory representation for all decoder decisions. We apply Gated Recurrent Unit (GRU) as the unit cell (Cho et al., 2014). We also compare the influence on performance of the $\lambda_{gen}$ parameter.

## 5 Experiments

### 5.1 Experimental Setup

**Baseline models** For the baseline models, we use the 2048-dimensional outputs of Google-Inception-v3 (Szegedy et al., 2015) (pre-trained on ImageNet ILSSVR 2012) to represent the images, and 1024-dimensional paragraph-vector embeddings (section 3.2) to represent captions. To reduce computation time, both are reduced to 256-dimensional vectors using random projections.

**Neural-nets based models** The experiments with these models are done using the Tensorflow package (Abadi et al., 2015). The hyper-parameter choices are decided using the hold-out development portion of the $\text{MC}_{\text{IC}}$ set. For modeling the input tokens, we use a vocabulary size of 8,855 types, selected as the most frequent tokens over the captions from the COCO training set (words occurring at least 5 times). The models are optimized using ADAGRAD with an initial learning rate of 0.01, and clipped gradients (maximum norm 4). We run the training procedures for $3,000,000$ steps, with a mini-batch size of 20. We use 40 workers for computing the updates, and 10 parameter servers for model storing and (asynchronous and distributed) updating.

We use the following notations to refer to the neural network models: $\text{FFNN}_{\text{2-class}}^{\text{argmax 1..5}}$ refers to the version of feedforward neural network architecture with a 2-class–classifier ('yes' or 'no' for answer correctness), over which an $\text{argmax}$ function computes a 5-way decision (i.e., the choice with the highest 'yes' probability); we henceforth refer to this model simply as FFNN.

The Vec2seq+FFNN refers to the hybrid model combining Vec2seq and $\text{FFNN}_{\text{2-class}}^{\text{argmax 1..5}}$. The RNN part of the model uses a two-hidden–layer GRU unit-cell (Cho et al., 2014) configuration, while the FFNN part uses a two-hidden–layer architecture. The $\lambda_{gen}$ hyper-parameter from the loss-function $L(\boldsymbol{\theta}, \mathbf{w}, \mathbf{u})$ (Eq. 4) is by default set to 1.0 (except for Section 5.3 where we directly measure its effect on performance).

We also include the result of a Vec2Seq model that is trained for caption generation only. To use the model for classification, we feed both image and each of its caption candidates to the Vec2Seq model, and then pick the caption candidate which has the lowest perplexity.

**Evaluation metrics** The metrics we use to measure performance come in two flavors. First, the accuracy in detecting (the index of) the true target among the decoys provides a direct way of measuring the performance level on the comprehension task. We use this metric as the main indicator of comprehension performance. Second, because our Vec2seq+FFNN models are multi-task models, they can also generate new captions given the input image. The performance level for the generation task is measured using the standard scripts measuring ROUGE-L (Lin and Och, 2004) and CIDEr (Vedantam et al., 2015), using as reference the available captions from the COCO data (around 5 for most of the images). Code for these metrics is available as part of the COCO evaluation toolkit [‡]. As usual, both the hypothesis strings and the reference strings are preprocessed: remove all the non-alphabetic characters; transform all letters to lowercase, and tokenize using white space; replace all words occurring less than 5 times with an unknown token $\langle \text{UNK} \rangle$ (total vocabulary of 8,855 types); truncate to the first 30 tokens.

### 5.2 Results

Table 2 summarizes our main results on the comprehension task. We report the accuracies (and their standard deviations) for random choice, baselines, and neural network-based models.

Interestingly, the Baseline-I2C model performs at the level of random choice, and much worse than the Baseline-C2I model. This discrepancy reflects the inherent difficulty in vision-

---

[‡] https://github.com/tylin/coco-caption

| Model | dim | Dev | Test |
|---|---|---|---|
| Baseline-I2C | 256 | 19.6 ±0.4 | 19.3±0.4 |
| Baseline-C2I | 256 | 32.8 ±0.5 | 32.0±0.5 |
| Baseline-LinM | 256 | 44.6 ±0.5 | 44.5±0.5 |
| FFNN | 256 | 56.3 ±0.5 | 55.1±0.5 |
| Vec2seq+FFNN | 256 | **60.5** ±0.5 | **59.0**±0.5 |
| Vec2Seq | 256 | 15.6 ± 0.4 | 16.0 ± 0.4 |

Table 2: Performance on the DMC Task, in accuracies (and standard deviations) on $MC_{IC}$ for baselines and NN models.

Language tasks: for each image, there are several possible equally good descriptions, thus a linear mapping from the image embeddings to the captions might not be enough – statistically, the *linear* model will just predict the mean of those captions. However, for the reverse direction where the captions are the independent variables, the learned model does not have to capture the variability in image embeddings corresponding to the different but equally good captions – there is only one such image embedding.

Nonlinear neural networks overcome these modeling limitations. The results clearly indicate their superiority over the baselines. The Vec2seq+FFNN model obtains the best results, with accuracies of 60.5% (dev) and 59.0% (test); the accuracy numbers indicate that the Vec2seq+FFNN architecture is superior to the non-recursive fully-connected FFNN architecture (at 55.1% accuracy on test). In the Suppl. Material, we show the impact on performance of the mebedding dimension and neural-network sizes.

Last but not least, the Vec2Seq model performs the worst in the classification task. This result indicates that a caption generation model alone is unable to discriminate among captions that are close in a pretrained embedding space, even when it has a good caption generation performance (CIDEr 0.983 on dev and 0.927 on test).

### 5.3 DMC and Image Captioning

In this section, we compare models with different values of $\lambda_{gen}$ in Eq. 4. This parameter allows for a natural progression from learning for the DMC task only ($\lambda_{gen} = 0$) to focusing more on the image captioning loss ($\lambda_{gen} = 16$).

The results in Table 3 illustrate one of the main points of this paper. That is, the ability to perform the comprehension task (as measured by the accuracy metric) positively correlates with the ability to perform other tasks that require machine comprehension, such as caption generation. At

| $\lambda_{gen}$ | Acc | | ROUGE-L | | CIDEr | |
|---|---|---|---|---|---|---|
| | Dev | Test | Dev | Test | Dev | Test |
| 0.0 | 50.7 | 50.7 ±0.5 | - | - | - | - |
| 0.1 | 61.1 | 59.0 ±0.5 | 0.517 | 0.511 | 0.901 | 0.865 |
| 1.0 | **63.4** | 60.8 ±0.5 | 0.528 | 0.518 | 0.972 | 0.903 |
| 2.0 | **63.4** | **61.3** ±0.5 | 0.528 | 0.519 | 0.971 | 0.921 |
| 4.0 | 63.0 | 60.9 ±0.5 | **0.533** | **0.524** | **0.989** | **0.938** |
| 8.0 | 62.1 | 60.1 ±0.5 | 0.526 | 0.520 | 0.957 | 0.914 |
| 16.0 | 61.8 | 59.6 ±0.5 | 0.530 | 0.519 | 0.965 | 0.912 |

Table 3: The impact of $\lambda_{gen}$ on $MC_{IC}$ accuracy, and caption-generation (ROUGE-L and CIDEr against 5 references). All results are obtained with a Vec2seq+FFNN model (embedding size 2048 and hidden-layer sizes of 2048 and 512).

$\lambda_{gen} = 4$, the Vec2seq+FFNN model not only has a high accuracy of detecting the ground-truth option, but it also generates its own captions given the input image, with an accuracy measured on $MC_{IC}$ at 0.9890 (dev) and 0.9380 (test) CIDEr scores. On the other hand, at an accuracy level of about 59% (on test, at $\lambda_{gen} = 0.1$), the generation performance is at only 0.9010 (dev) and 0.8650 (test) CIDEr scores.

We note that there is an inherent trade-off between prediction accuracy and generation performance, as seen for $\lambda_{gen}$ values above 4.0. This agrees with the intuition that training a Vec2seq+FFNN model using a loss $L(\boldsymbol{\theta}, \mathbf{w}, \mathbf{u})$ with a larger $\lambda_{gen}$ means that the ground-truth detection loss (the first term of the loss in Eq.4) may get overwhelmed by the word-generation loss (the second term). However, our empirical results suggest that there is value in training models with a multi-task setup, in which both the comprehension side as well as the generation side are carefully tuned to maximize performance.

## 6 Discussion

We have proposed and described in detail a new multi-modal machine comprehension task (DMC). The underlying hypothesis is that computer systems that can be shown to perform increasingly well on this task will do so by constructing a visually-grounded understanding of various linguistic elements and their dependencies.

The Vec2seq+FFNN architecture can be trained end-to-end to display both the ability to choose the most likely text associated with an image, as well as the ability to generate a complex description of that image. The empirical results validate that improvements in comprehension and generation happen in tandem.

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
