# Peer review of "Understanding Image and Text Simultaneously: a Dual Vision-Language Machine Comprehension Task"

_ACL 2017 — decision unknown_

[Official Review · Reviewer 1 · rating 2 · confidence 4]
soundness 5 · originality 5 · clarity 5 · impact 3 · substance 4 · appropriateness 5 · meaningful comparison 3 · presentation format Poster

The paper proposes a task of selecting the most appropriate textual description
for a given scene/image from a list of similar options. It also proposes couple
of baseline models, an evaluation metrics and human evaluation score. 

- Strengths:

The paper is well-written and well-structured. 
It is clear with its contributions and well supports them by empirical
evidence. So the paper is very easy to read. 

The paper is well motivated. A method of selecting the most appropriate caption
given a list of misleading candidates will benefit other
image-caption/understanding models, by acting as a post-generation re-ranking
method. 

- Weaknesses:

I am not sure if the proposed algorithm for decoys generation is effective,
which as a consequence puts the paper on questions.

For each target caption, the algorithm basically picks out those with similar
representation and surface form but do not belong to the same image. But a
fundamentally issue with this approach is: not belonging to the image-A does
not mean not appropriate to describe image-A, especially when the
representation and surface form are close. So the ground-truth labels might not
be valid. As we can see in Figure-1, the generated decoys are either too far
from the target to be a *good* decoy (*giraffe* vs *elephant*), or fair
substitutes for the target (*small boy playing kites* vs *boy flies a kite*).

Thus, I am afraid that the dataset generated with this algorithm can not train
a model to really *go beyond key word recognition*, which was claimed as
contribution in this paper. As shown in Figure-1, most
decoys can be filtered by key word mismatch---*giraffe vs elephant*, *pan vs
bread*, *frisbee vs kite*, etc. And when they can not be separated by *key word
match*, they look very tempting to be a correct option.

Furthermore, it is interesting that humans only do correctly on 82.8% on a
sampled test set. Does it mean that those examples are really too hard even for
human to correctly classify? Or are some of the *decoys* in fact good enough to
be the target's substitute (or even better) so that human choose them over
ground-truth targets?

- General Discussion:

I think this is a well-written paper with clear motivation and substantial
experiments. 
The major issue is that the data-generating algorithm and the generated dataset
do not seem helpful for the motivation. This in turn makes the experimental
conclusions less convincing. So I tend to reject this paper unless my concerns
can be fully addressed in rebuttal.

[Official Review · Reviewer 2 · rating 3 · confidence 3]
soundness 5 · originality 5 · clarity 3 · impact 3 · substance 3 · appropriateness 5 · meaningful comparison 3 · presentation format Poster

- Strengths:

Authors generate a dataset of “rephrased” captions and are planning to make
this dataset publicly available.

The way authors approached DMC task has an advantage over VQA or caption
generation in terms of metrics. It is easier and more straightforward to
evaluate problem of choosing the best caption. Authors use accuracy metric.
While for instance caption generation requires metrics like BLUE or Meteor
which are limited in handling semantic similarity.

Authors propose an interesting approach to “rephrasing”, e.g. selecting
decoys. They draw decoys form image-caption dataset. E.g. decoys for a single
image come from captions for other images. These decoys however are similar to
each other both in terms of surface (bleu score) and semantics (PV similarity).
Authors use lambda factor to decide on the balance between these two components
of the similarity score. I think it would be interesting to employ these for
paraphrasing.

Authors support their motivation for the task with evaluation results. They
show that a system trained with the focus on differentiating between similar
captions performs better than a system that is trained to generate captions
only. These are, however, showing that system that is tuned for a particular
task performs better on this task.

- Weaknesses:

 It is not clear why image caption task is not suitable for comprehension task
and why author’s system is better for this. In order to argue that system can
comprehend image and sentence semantics better one should apply learned
representation, e.g. embeddings. E.g. apply representations learned by
different systems on the same task for comparison.

My main worry about the paper is that essentially authors converge to using
existing caption generation techniques, e.g. Bahdanau et al., Chen et al.

They way formula (4) is presented is a bit confusing. From formula it seems
that both decoy and true captions are employed for both loss terms. However, as
it makes sense, authors mention that they do not use decoy for the second term.
That would hurt mode performance as model would learn to generate decoys as
well. The way it is written in the text is ambiguous, so I would make it more
clear either in the formula itself or in the text. Otherwise it makes sense for
the model to learn to generate only true captions while learning to distinguish
between true caption and a decoy.

- General Discussion:

Authors formulate a task of Dual Machine Comprehension. They aim to accomplish
the task by challenging computer system to solve a problem of choosing between
two very similar captions for a given image. Authors argue that a system that
is able to solve this problem has to “understand” the image and captions
beyond just keywords but also capture semantics of captions and their alignment
with image semantics.

I think paper need to make more focus on why chosen approach is better than
just caption generation and why in their opinion caption generation is less
challenging for learning image and text representation and their alignment.

For formula (4). I wonder if in the future it is possible to make model to
learn “not to generate” decoys by adjusting second loss term to include
decoys but with a negative sign. Did authors try something similar?

[Official Review · Reviewer 3 · rating 2 · confidence 5]
soundness 5 · originality 5 · clarity 3 · impact 3 · substance 3 · appropriateness 5 · meaningful comparison 3 · presentation format Oral Presentation

- Strengths:

The DMC task seems like a good test of understanding language and vision. I
like that the task has a clear evaluation metric.

The failure of the caption generation model on the DMC task is quite
interesting. This result further demonstrates that these models are good
language models, but not as good at capturing the semantics of the image.

- Weaknesses:

The experiments are missing a key baseline: a state-of-the-art VQA model
trained with only a yes/no label vocabulary. 

I would have liked more details on the human performance experiments. How many
of the ~20% of incorrectly-predicted images are because the captions are
genuinely ambiguous? Could the data be further cleaned up to yield an even
higher human accuracy?

- General Discussion:

My concern with this paper is that the data set may prove to be easy or
gameable in some way. The authors can address this concern by running a suite
of strong baselines on their data set and demonstrating their accuracies. I'm
not convinced by the current set of experiments because the chosen neural
network architectures appear quite different from the state-of-the-art
architectures in similar tasks, which typically rely on attention mechanisms
over the image.

Another nice addition to this paper would be an analysis of the data set. How
many tokens does the correct caption share with distractors on average? What
kind of understanding is necessary to distinguish between the correct and
incorrect captions? I think this kind of analysis really helps the reader
understand why this task is worthwhile relative to the many other similar
tasks. 

The data generation technique is quite simple and wouldn't really qualify as a
significant contribution, unless it worked surprisingly well.

- Notes

I couldn't find a description of the FFNN architecture in either the paper or
the supplementary material. It looks like some kind of convolutional network
over the tokens, but the details are very unclear. I'm also confused about how
the Veq2Seq+FFNN model is applied to both classification and caption
generation. Is the loglikelihood of the caption combined with the FFNN
prediction during classification? Is the FFNN score incorporated during caption
generation?

The fact that the caption generation model performs (statistically
significantly) *worse* than random chance needs some explanation. How is this
possible?

528 - this description of the neural network is hard to understand. The final
paragraph of the section makes it clear, however. Consider starting the section
with it.